# SAMPLE AND COMPUTATION REDISTRIBUTION FOR EFFICIENT FACE DETECTION

**Jia Guo[2], Jiankang Deng[1,2] \*, Alexandros Lattas[1,3], Stefanos Zafeiriou[1,3]**
[1]Huawei, [2]InsightFace, [3]Imperial College London
{guojia,jiankangdeng}@gmail.com, {a.lattas,s.zafeiriou}@imperial.ac.uk

## ABSTRACT

Although tremendous strides have been made in uncontrolled face detection, accurate face detection with a low computation cost remains an open challenge. In this paper, we point out that computation distribution and scale augmentation are the keys to detecting small faces from low-resolution images. Motivated by these observations, we introduce two simple but effective methods: (1) Computation Redistribution (CR), which reallocates the computation between the backbone, neck and head of the model; and (2) Sample Redistribution (SR), which augments training samples for the most needed stages. The proposed Sample and Computation Redistribution for Face Detection (SCRFD) is implemented by a random search in a meticulously designed search space. Extensive experiments conducted on WIDER FACE demonstrate the state-of-the-art accuracy-efficiency trade-off for the proposed SCRFD family across a wide range of compute regimes. In particular, SCRFD-34GF outperforms the best competitor, TinaFace, by $4.78\%$ (AP at hard set) while being more than *3× faster* on GPUs with VGA-resolution images. Code is available at: https://github.com/deepinsight/insightface/tree/master/detection/scrfd.

## 1    INTRODUCTION

Face detection is a long-standing problem in computer vision with many applications, such as face alignment (Bulat & Tzimiropoulos, 2017; Deng et al., 2019b), face reconstruction (Feng et al., 2018; Gecer et al., 2021), face attribute analysis (Zhang et al., 2018; Pan et al., 2018), and face recognition (Schroff et al., 2015; Deng et al., 2019a; 2020a). Following the pioneering work of (Viola & Jones, 2004), numerous face detection algorithms have been designed. Among them, the single-shot anchor-based approaches (Najibi et al., 2017; Zhang et al., 2017b; Tang et al., 2018; Li et al., 2019; Ming et al., 2019; Deng et al., 2020b; Liu et al., 2020; Zhu et al., 2020) have recently demonstrated very promising performance. In particular, on the most challenging face detection dataset, WIDER FACE (Yang et al., 2016), the average precision (AP) on its hard validation set has been boosted to $93.4\%$ by TinaFace (Zhu et al., 2020).

Even though TinaFace (Zhu et al., 2020) achieves impressive results on unconstrained face detection, it employs large-scale (*e.g.* $1,650$ pixels) testing, which consumes huge amounts of computational resources. In addition, TinaFace design is based on a generic object detector (*i.e.* RetinaNet (Lin et al., 2017b)), directly taking the classification network as the backbone, tiling dense anchors on the multi-scale feature maps (*i.e.* P2 to P7 of neck), and adopting heavy head designs. Without considering the prior of faces, the network design of TinaFace is thus redundant and sub-optimal.

One approach of optimizing such networks' performance is computation redistribution. Since directly taking the backbone of the classification network for object detection is sub-optimal, the recent CR-NAS (Liang et al., 2020) reallocates the computation across different resolutions to obtain a more balanced Effective Receptive Field (ERF), leading to higher detection performance. In BFbox (Liu & Tang, 2020), a face-appropriate search space is designed, based on the observation of scale distribution gap between general object detection and face detection. In ASFD (Zhang et al.,

---

*denotes equal contribution and corresponding author. InsightFace is a nonprofit Github project for 2D and 3D face analysis.

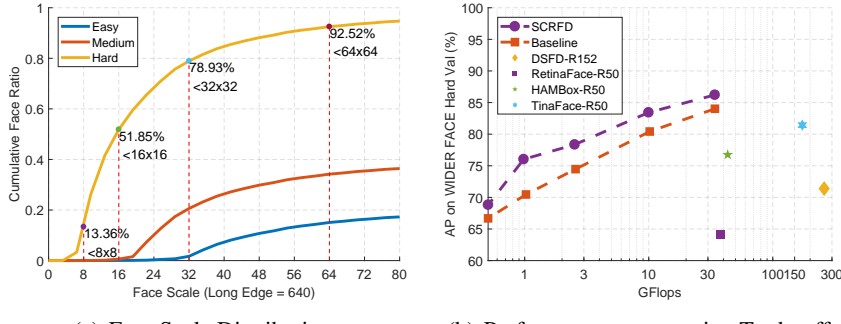

(a) Face Scale Distribution      (b) Performance-computation Trade-off

Figure 1: (a) Cumulative face scale distribution on the WIDER FACE validation dataset (Easy $\subset$ Medium $\subset$ Hard). When the long edge is fixed as $640$ pixels, most of the easy faces are larger than $32 \times 32$, and most of the medium faces are larger than $16 \times 16$. For the hard track, 78.93% faces are smaller than $32 \times 32$, 51.85% faces are smaller than $16 \times 16$, and 13.36% faces are smaller than $8 \times 8$. (b) Performance-computation trade-off on the WIDER FACE validation hard set for different face detectors. Flops and APs are reported by using the VGA resolution ($640 \times 480$) during testing. The proposed SCRFD outperforms a range of state-of-the-art open-sourced methods by using much fewer flops.

2020a), a differential architecture search is employed to discover optimized feature enhance modules for efficient multi-scale feature fusion and context enhancement. Even though (Liu & Tang, 2020; Zhang et al., 2020a) have realized the limitation of directly applying general backbone, neck and head settings to face detection, CR-NAS (Liang et al., 2020) only focuses the optimization on backbone, BFbox (Liu & Tang, 2020) neglects the optimization of head, and ASFD (Zhang et al., 2020a) only explores the best design for neck.

Another optimization approach, is the sample redistribution across different scales. Due to the extremely large scale variance of faces in real-world scenarios, different scale augmentation strategies are employed to introduce scale adaptation into the face detector. The most widely used scale augmentation approaches include random square crop (Zhang et al., 2017b; Deng et al., 2020b; Zhu et al., 2020) and data anchor sampling (Tang et al., 2018). Nevertheless, the scale augmentation parameters in these methods are manually designed for all different network structures. Therefore, traditional multi-scale training in face detection is also tedious and sub-optimal.

Since VGA resolution ($640 \times 480$) is widely used for efficient face detection on numerous mobile phones and digital cameras, we focus on efficient face detection from low-resolution images in this paper. In Fig 1(a), we give the cumulative face scale distribution on the WIDER FACE validation dataset. Under the VGA resolution, most of the faces (78.93%) in WIDER FACE are smaller than $32 \times 32$ pixels. Under this specific scale distribution, both network structure and scale augmentation need to be optimized.

In this work, we present a meticulously designed methodology of search space optimization, that addresses both the redistribution between the backbone, neck and head, and the sample redistribution between the most needed scales. As the structure of a face detector determines the distribution of computation and is the key in determining its accuracy and efficiency, we first discover principles of computation distribution under different flop regimes. Inspired by (Radosavovic et al., 2020), we control the degrees of freedom and reduce the search space. More specifically, we randomly sample model architectures with different configurations on backbone (stem and four stages), neck and head. Based on the statistics of these models, we compute the empirical bootstrap (Efron & Tibshirani, 1994) and estimate the likely range in which the best models fall. To further decrease the complexity of the search space, we divide the computation ratio estimation for backbone and the whole detector into two steps. To handle extreme scale variations in face detection, we also design a search-able zoom-in and zoom-out space, specified by discrete scales and binary probabilities. In experiments, the proposed computation redistribution and sample redistribution yield significant and consistent improvement on various compute regimes, even surpassing a range of state-of-the-art face detectors by using much fewer flops as shown in Fig. 1(b).

To sum up, this paper makes following contributions:

- We have proposed a simplified search space, as well as a two-step search strategy for computation redistribution across different components (backbone, neck and head) of a face detector. The proposed computation redistribution method can easily boost detection performance through random search.
- We have designed a search-able zoom-in and zoom-out space for face-specific scale augmentation, which automatically redistributes more training samples for shallow stages, enhancing the detection performance on small faces.
- Extensive experiments conducted on WIDER FACE demonstrate the significantly improved accuracy and efficiency trade-off of the proposed SCRFD across a wide range of compute regimes.

## 2 RELATED WORK

**Face Detection.** To deal with extreme variations (*e.g.* scale, pose, illumination and occlusion) in face detection (Yang et al., 2016), most of the recent single-shot face detectors focus on improving the anchor sampling/matching or feature enhancement. SSH (Najibi et al., 2017) builds detection modules on different feature maps with a rich receptive field. $S^3FD$ (Zhang et al., 2017b) introduces an anchor compensation strategy by offsetting anchors for outer faces. PyramidBox (Tang et al., 2018) formulates a data-anchor-sampling strategy to increase the proportion of small faces in the training data. DSFD (Li et al., 2019) introduces small faces supervision signals on the backbone, which implicitly boosts the performance of pyramid features. Group sampling (Ming et al., 2019) emphasizes the importance of the ratio for matched and unmatched anchors. RetinaFace (Deng et al., 2020b) employs deform-able context modules and additional landmark annotations to improve the performance of face detection. HAMBox (Liu et al., 2020) finds that many unmatched anchors in the training phase also have strong localization ability and proposes an online high-quality anchor mining strategy to assign high-quality anchors for outer faces. BFbox (Liu & Tang, 2020) employs a single-path one-shot search method (Guo et al., 2019) to jointly optimize the backbone and neck for face detector. ASFD (Zhang et al., 2020a) explores a differential architecture search to discover optimized feature enhance modules for efficient multi-scale feature fusion and context enhancement. All these methods are either designed by expert experience or partially optimized on backbone, neck and head. By contrast, we search for computation redistribution across different components (backbone, neck and head) of a face detector across a wide range of compute regimes.

**Neural Architecture Search.** Given a fixed search space of possible networks, Neural Architecture Search (NAS) automatically finds a good model within the search space. DetNAS (Chen et al., 2019b) adopts the evolution algorithm for the backbone search to boost object detection on COCO (Lin et al., 2014). By contrast, CR-NAS (Liang et al., 2020) reallocates the computation across different stages within the backbone to improve object detection. NAS-FPN (Ghiasi et al., 2019) uses reinforcement learning to search the proper FPN for general object detection. As there is an obvious distribution gap between COCO (Lin et al., 2014) and WIDER FACE (Yang et al., 2016), the experience in the above methods is not directly applicable for face detection but gives us an inspiration that the backbone, neck and head can be optimized to enhance the performance of face detection. Inspired by RegNet (Radosavovic et al., 2020), we optimize the computation distribution on backbone, neck and head based on the statistics from a group of random sampled models. We successfully reduce the search space and find the stable computation distribution under a particular complex regime, which significantly improves the model's performance.

## 3 METHODOLOGY

To efficiently and accurately detect small faces from low-resolution images (*e.g.* VGA $640 \times 480$), we propose two methodologies that, when combined, outperform the state-of-the-art. In Sec. 3.1, we explore the computation redistribution across different stages of backbone, as well as different components (*i.e.* backbone, neck and head) of the whole detector, given a pre-defined computation budget. Then, in Sec. 3.2, we investigate the redistribution of positive training samples across different scales of feature maps by searching optimized scale augmentations.

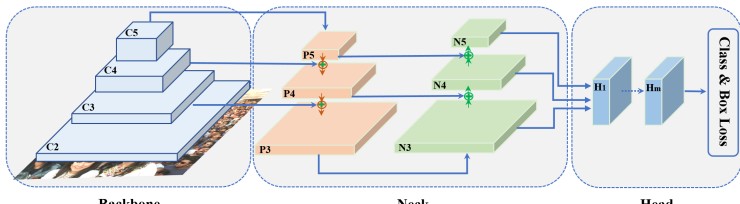

Figure 2: Computation redistribution among the backbone, neck and head. The backbone search space contains four stages, each stage having two parameters: the block number $d_i$ and block width $w_i$. The neck search space only includes the channel number $n$. The head is shared for the three-scale of feature maps ($N_i$), and the search space consists of the block number $m$ and channel number $h$.

## 3.1 COMPUTATION REDISTRIBUTION

As illustrated in Fig. 2, we apply our search method on a network consisting of (1) RetinaNet (Lin et al., 2017a), with ResNet (He et al., 2016) as the backbone, (2) Path Aggregation Feature Pyramid Network (PAFPN) (Liu et al., 2018) as the neck, and (3) stacked $3 \times 3$ convolutional layers for the head. Despite the generally simple structure, the total number of possible network configurations of the search space becomes unwieldy. Therefore, we attempt to simplify the tremendous search space and arrive at a low-dimensional design space, consisting of simple and effective networks.

### 3.1.1 SEARCH SPACE DESIGN

Inspired by RegNet (Radosavovic et al., 2020), we explore the structures of face detectors, assuming fixed standard network blocks (*i.e.*, basic residual or bottleneck blocks with a fixed bottleneck ratio of 4). In our case, the structure of a face detector includes:

- the *backbone stem*, three $3 \times 3$ convolutional layers with $w_1$ output channels (He et al., 2019a).
- the *backbone body*, four stages (*i.e.* C2, C3, C4 and C5) operating at progressively reduced resolution, with each stage consisting of a sequence of identical blocks. For each stage $i$, the degrees of freedom include the number of blocks $d_i$ (*i.e.* network depth) and the block width $w_i$ (*i.e.* number of channels).
- the *neck*, a multi-scale feature aggregation module by a top-down path and a bottom-up path with $n$ channels (Liu et al., 2018).
- the *head*, with $h_i$ channels of $m$ blocks to predict face scores and regress face boxes.

The search space can be initially designed as follows. As the channel number of the stem is equal to the block width of the first residual block in C2, the degree of freedom of the stem $w_1$ can be merged into $w_2$. In addition, we employ a shared head design for three-scale of feature maps and fix the channel number for all $3 \times 3$ convolutional layers within the heads. Therefore, we reduce the degrees of freedom to three within our neck and head design: (1) output channel number $n$ for neck, (2) output channel number $h$ for head, and (3) the number of $3 \times 3$ convolutional layers $m$. We perform uniform sampling of $n \leq 256$, $h \leq 256$, and $m \leq 6$ (both $n$ and $h$ are divisible by 8).

The backbone search space has 8 degrees of freedom as there are 4 stages and each stage $i$ has 2 parameters: the number of blocks $d_i$ and block width $w_i$. Following RegNet (Radosavovic et al., 2020), we perform uniform sampling of $d_i \leq 24$ and $w_i \leq 512$ ($w_i$ is divisible by 8). As state-of-the-art backbones have increasing widths (Radosavovic et al., 2020), we also constrain the search space, according to the principle of $w_{i+1} \geq w_i$.

### 3.1.2 ESTIMATION METRIC

Based on above simplifications, our search space becomes more compact and efficient. We repeat the random sampling in our search space until we obtain 320 models in our target complexity regime, and train each model on the WIDER FACE (Yang et al., 2016) training set for 80 epochs. Then, we test the Average Precision (AP) of each model on the validation set. Based on these 320 pairs of model statistics $(x_i, AP_i)$, where $x_i$ is the computation ratio of a particular component and $AP_i$ the corresponding performance, we can compute the empirical bootstrap (Efron & Tibshirani, 1994) to estimate the likely range in which the best models fall. More specifically, we repeatedly sample with replacement $25\%$ of the pairs for $10^3$ times and select the pair with maximum AP in each sampling. Afterwards, we compute the $95\%$ confidence interval for the maximum value and the median gives the most likely best computation ratio.

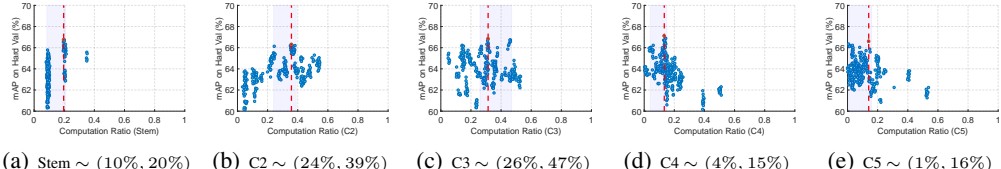

(a) Stem ~ (10%, 20%)  (b) C2 ~ (24%, 39%)  (c) C3 ~ (26%, 47%)  (d) C4 ~ (4%, 15%)  (e) C5 ~ (1%, 16%)

Figure 3: Computation redistribution on the backbone (stem, C2, C3, C4 and C5) with fixed neck and head under the constraint of 2.5 GFlops. For each component within the backbone, the range of computation ratio in which the best models may fall is estimated by the empirical bootstrap.

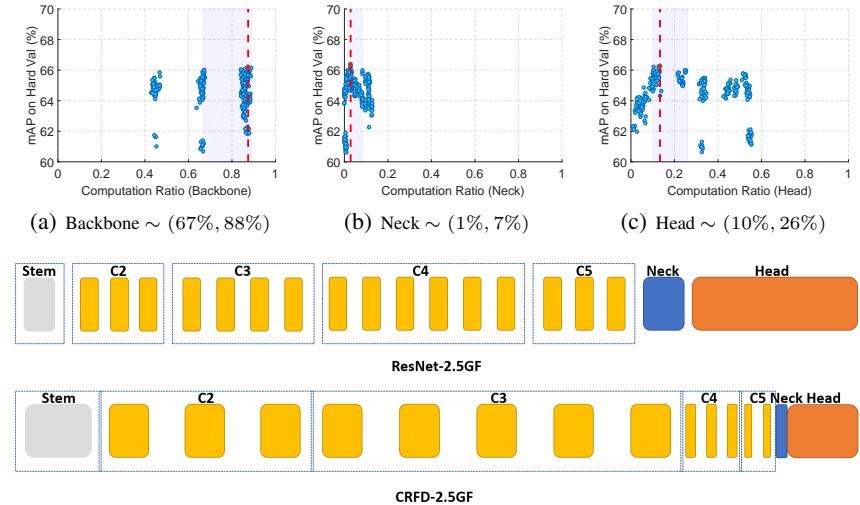

(a) Backbone ~ (67%, 88%)  (b) Neck ~ (1%, 7%)  (c) Head ~ (10%, 26%)

(d) Architecture Sketches

Figure 4: Computation redistribution and architecture sketches under the constraint of 2.5 GFlops. The computation distribution of CRFD-2.5GF within backbone follows Fig. 3. In (d), the yellow rectangles in C2 to C5 represents the basic residual block. The width of rectangles corresponds to the computation cost. After computation redistribution, more computations are allocated to shallow stages (*i.e.* C2 and C3).

### 3.1.3 TWO-STEP SEARCH

To further decrease the complexity of search space, we divide our network structure search into the following two steps: (1) CRFD$_1$: search the computational distribution for the backbone only, while fixing the settings of the neck and head to the default configuration, and (2) CRFD$_2$: search the computational distribution over the whole face detector (*i.e.* backbone, neck and head), with the computational distribution within the backbone, following the optimized CRFD$_1$. By optimizing in both manners, we achieve the final optimized network design for the computation-constrained face detection. In the example below, we constrain CRFD to 2.5 GFlops (CRFD-2.5GF), in order to illustrate our two-step searching strategy.

**Computation redistribution on backbone.** For CRFD$_1$-2.5GF, we fix the output channel of the neck at 32 and use two stacked $3 \times 3$ convolutions with 96 output channels. As the neck and head configurations do not change in the whole search process of CRFD$_1$, we can easily find the best computation distribution of the backbone. As described in Fig. 3, we show the distribution of 320 model APs (on the WIDER FACE hard validation set) versus the computation ratio over each component (*i.e.* stem, C2, C3, C4 and C5) of backbone. After applying an empirical bootstrap (Efron & Tibshirani, 1994), a clear trend emerges, showing that the backbone computation is reallocated to the shallow stages (*i.e.* C2 and C3).

**Computation redistribution on backbone, neck and head.** In this step, we only keep the randomly generated network configurations whose backbone settings follow the computation distribution from CRFD$_1$ as shown in Fig. 3. In this case, there are another three degrees of freedom (*i.e.* output channel number $n$ for neck, output channel number $h$ for head, and the number $m$ of $3 \times 3$ convolutional layers in head). We repeat the random sampling in our search space, until we obtain 320 qualifying models in our target complexity regime (*i.e.* 2.5 GFlops). As evident in Fig. 4, most

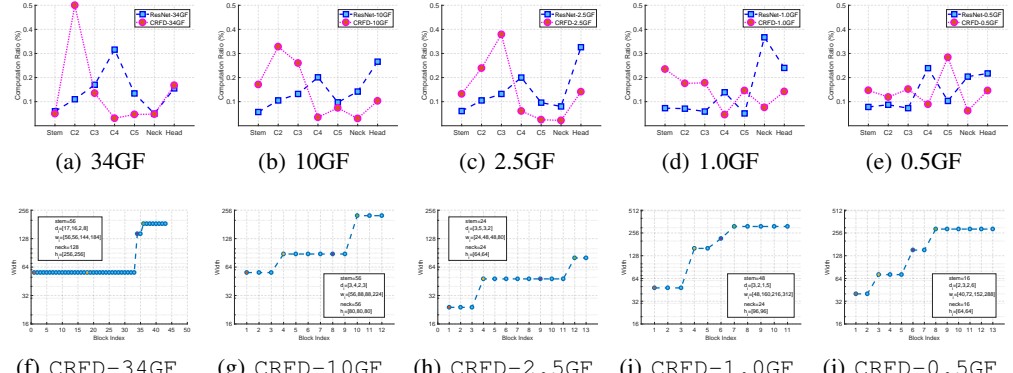

(a) 34GF  (b) 10GF  (c) 2.5GF  (d) 1.0GF  (e) 0.5GF

(f) CRFD-34GF  (g) CRFD-10GF  (h) CRFD-2.5GF  (i) CRFD-1.0GF  (j) CRFD-0.5GF

Figure 5: Computation redistribution and the searched network structures under different computation constraints. Network diagram legends in the second row contain all information required to implement the CRFD models that we have optimized the computation across stages and components.

of the computation is allocated in the backbone, with the head following and the neck having the lowest computation ratio. Fig. 4(d) also depicts the comparison between the hand-crafted model architecture and the computation redistributed network, under the constraint of 2.5 GFlops.

## 3.2 SAMPLE REDISTRIBUTION

As face detection features large scale variations (from several pixels to thousand pixels), there exist two widely used scale augmentation strategies, random square crop (Zhang et al., 2017b; Deng et al., 2020b; Zhu et al., 2020) and data anchor sampling (Tang et al., 2018). In the random square crop strategy, square patches are cropped from the original image with a random size between $[0.3, 1]$ of the short edge and then resized into $640 \times 640$ to generate larger training faces. By contrast, data anchor sampling strategy aims to generate more small scale faces by down-sampling the original image, bringing a large amount of padded area. Even though both random square crop and data anchor sampling can achieve promising results on the WIDER FACE dataset, the scale augmentation parameters are manually designed for all different network structures. Therefore, the training sample distribution on the feature pyramids can be sub-optimal for a particular network structure.

To handle extreme scale variations in face detection, we also design a search-able zoom-in and zoom-out space, specified by the scale $s_i$ and probability $p_i$. The scale $s_i$ represents the zooming ratio, sampled from a discrete set $\mathbb{S} = \{s_{min}, s_{min} + 0.1, \cdots, s_{max} - 0.1, s_{max}\}$. For a particular training image in each iteration, square patches are cropped from the original images with a zooming ratio $s_i$ of the short edge of the original images. If the square patch is larger than the original image, average RGB values will fill the missing pixels. To shrink the scale search space, we employ a binary probability set $p_i \in \{0, 1\}$. Under this setting, the probability-based scale search is simplified into a discrete scale sampling from a fixed set. As the interval of the discrete scale set is only $0.1$, adjacent scales will have the probability of $1.0$ to approximate a higher probability around a particular scale. In this paper, we employ random search under the estimation metric of AP on WIDER FACE to construct the best scale augmentation set. More specifically, we set $s_{min} = 0.1$ and $s_{max} = 3.0$. Then, we randomly select $8$ to $20$ discrete scale values to construct each scale augmentation set and train CRFD models under $320$ different scale augmentation sets. Finally, the scale augmentation set with the highest detection performance is selected for optimized scale augmentation.

## 4 EXPERIMENTS

### 4.1 IMPLEMENTATION DETAILS

**Training.** For the scale augmentation, square patches are cropped from the original images with a random size from a pre-defined scale set, and then these patches are resized to $640 \times 640$ for training. Besides scale augmentation, the training data are also augmented by color distortion and random horizontal flipping, with a probability of $0.5$. For the anchor setting, we tile anchors of $\{16, 32\}$, $\{64, 128\}$, and $\{256, 512\}$ on the feature maps of stride 8, 16, and 32, respectively. The anchor ratio is set as $1.0$. In this paper, we employ Adaptive Training Sample Selection (ATSS)

Table 1: Ablation experiments of `SCRFD-2.5GF` (*i.e.* CR@two-steps+SR) on the WIDER FACE validation subset. "CR" and "SR" denote the proposed computation and sample redistribution, respectively. Results are reported on the single-scale VGA resolution.

| Method | Scale Augmentation Set | Easy | Medium | Hard |
|---|---|---|---|---|
| ResNet-2.5GF | [0.3, 1.0] | 91.87 | 89.49 | 67.32 |
| BFBox-2.5GF (Liu & Tang, 2020) | [0.3, 1.0] | 92.22 | 90.19 | 69.41 |
| Evolutionary-2.5GF | [0.3, 1.0] | 92.30 | 90.21 | 69.62 |
| CR@backbone | [0.3, 1.0] | 92.32 | 90.25 | 69.78 |
| CR@detector | [0.3, 1.0] | 92.61 | 90.74 | 70.98 |
| CR@two-steps (`CRFD-2.5GF`) | [0.3, 1.0] | 92.66 | 90.72 | 71.37 |
| ResNet-2.5GF | [0.3,2.0] | 93.21 | 91.11 | 74.47 |
| ResNet-2.5GF | SR | 93.17 | 91.14 | 74.93 |
| CR@two-steps | [0.3,2.0] | 93.78 | 92.16 | 77.87 |
| CR@two-steps (`SCRFD-2.5GF`) | SR | 93.76 | 92.17 | 78.35 |

(Zhang et al., 2020b) for positive anchor matching. In the detection head, weight sharing and Group Normalization (Wu & He, 2018) are used. The losses of classification and regression branches are Generalized Focal Loss (GFL) (Li et al., 2020) and DIoU loss (Zheng et al., 2020), respectively.

Our experiments are implemented in PyTorch, based on the open-source MMDetection (Chen et al., 2019a). We adopt the SGD optimizer (momentum 0.9, weight decay 5e-4) with a batch size of $8 \times 8$ and train on eight Tesla V100. The learning rate is linearly warmed up to $0.015$ within the first 3 epochs. During network search, the learning rate is multiplied by $0.1$ at the 55-th, and 68-th epochs. The learning process terminates on the 80-th epoch. For training of both baselines and searched configurations, the learning rate decays by a factor of 10 at the 440-th and 544-th epochs, and the learning process terminates at the 640-th epoch. All the models are trained from scratch without any pre-training.

**Testing.** For fair comparisons with other methods, we employ three testing strategies, including single-scale VGA resolution ($640 \times 480$), single-scale original resolution, and multi-scale testing. The results of DSFD (Li et al., 2019), RetinaFace (Deng et al., 2020b), TinaFace (Zhu et al., 2020), Faceboxes (Zhang et al., 2017a), libfacedetection (Feng et al., 2021) and LFFD (He et al., 2019b) are reported by testing the released models, while the HAMBox (Liu et al., 2020) and BFBox (Liu & Tang, 2020) models are shared from the author.

## 4.2 ABLATION STUDY

In Tab. 1, we present the performance of models on the WIDER FACE dataset by gradually including the proposed computation and sample redistribution methods. Our manually-designed baseline model, ResNet-2.5GF, gets APs of $91.87\%$, $89.49\%$, and $67.32\%$ under three validation scenarios.

**Computation redistribution.** After separately employing the proposed computation redistribution on the backbone and the whole detector, the AP on the hard set improves to $69.78\%$ and $70.98\%$. This indicates that (1) the network structure directly inherited from the classification task is suboptimal for the face detection task, and (2) joint computation reallocation on the backbone, neck and head outperforms computation optimization applied only on the backbone. Furthermore, the proposed two-step computation redistribution strategy achieves AP of $71.37\%$, surpassing one-step computation reallocation on the whole detector by $0.39\%$. As we shrink the whole search space by the proposed two-step strategy and our random model sampling number is fixed at 320, the two-step method is possible to find better network configurations from the large search space. In Tab. 1, we also compare our method with the single path one-shot NAS method (BFBox (Liu & Tang, 2020)) and the evolutionary search method (Appendix A.2), under the constraint of 2.5 GFlops. BFBox aims to design a face-appropriate search space by combing some excellent block designs, such as bottleneck block, densenet block and shufflenet block. However, such a combination generates a complex and redundant search space, which inevitably involves a vast body of low-performance candidate architectures. The evolutionary approach iteratively adopts mutations and crossover to gradually generate better architecture candidates from the randomly initialized search space, which also contains a large number of under-performing architectures. By contrast, `CRFD-2.5GF` utilizes an empirical bootstrap to estimate the optimized computation distribution of the best-performing architecture candidates, which directly eliminates the low-quality architectures from the initialized search

Table 2: Accuracy and efficiency of different methods on the WIDER FACE validation set. #Params and #Flops denote the number of parameters and multiply-adds. "Infer" refers to network inference latency on NVIDIA 2080TI.

| Method | Backbone | Easy | Medium | Hard | #Params(M) | #Flops(G) | Infer(ms) |
|--------|----------|------|--------|------|------------|-----------|-----------|
| DSFD@VGA | ResNet152 | 94.29 | 91.47 | 71.39 | 120.06 | 259.55 | 55.6 |
| DSFD@Multi-Scale | ResNet152 | 96.6 | 95.7 | 90.4 | 120.06 | 15928.5 | - |
| RetinaFace@VGA | ResNet50 | 94.92 | 91.90 | 64.17 | 29.50 | 37.59 | 21.7 |
| RetinaFace@Multi-Scale | ResNet50 | 96.7 | 96.1 | 91.4 | 29.50 | 4585.98 | - |
| BFBox@VGA | - | 94.2 | 92.1 | 70.4 | 28.6 | 39.4 | 22.4 |
| BFBox@Multi-Scale | - | 96.5 | 95.7 | 91.7 | 28.6 | 4732.8 | - |
| HAMBox@VGA | ResNet50 | 95.27 | 93.76 | 76.75 | 30.24 | 43.28 | 25.9 |
| HAMBox@Multi-Scale | ResNet50 | 97.0 | 96.4 | 93.3 | 30.24 | 5246.23 | - |
| TinaFace@VGA | ResNet50 | 95.61 | 94.25 | 81.43 | 37.98 | 172.95 | 38.9 |
| TinaFace@Multi-Scale | ResNet50 | 97.0 | 96.3 | 93.4 | 37.98 | 42333.64 | - |
| ResNet-34GF@VGA | ResNet50 | 95.64 | 94.22 | 84.02 | 24.81 | 34.16 | 11.8 |
| CRFD-34GF@VGA | Bottleneck Res | 96.06 | 94.92 | 85.29 | 9.80 | 34.13 | 11.7 |
| SCRFD-34GF@VGA | Bottleneck Res | 96.05 | 94.96 | 86.21 | 9.80 | 34.13 | 11.7 |
| SCRFD-34GF@Multi-Scale | Bottleneck Res | 97.20 | 96.58 | 93.53 | 9.80 | 2098.98 | - |
| ResNet-10GF@VGA | ResNet34x0.5 | 94.69 | 92.90 | 80.42 | 6.85 | 10.18 | 6.3 |
| CRFD-10GF@VGA | Basic Res | 95.16 | 93.87 | 83.05 | 3.86 | 9.98 | 4.9 |
| SCRFD-10GF@VGA | Basic Res | 95.14 | 93.96 | 83.43 | 3.86 | 9.98 | 4.9 |
| SCRFD-10GF@Multi-Scale | Basic Res | 95.93 | 94.95 | 90.81 | 3.86 | 614.14 | - |
| ResNet-2.5GF@VGA | ResNet34x0.25 | 93.21 | 91.11 | 74.47 | 1.62 | 2.57 | 5.4 |
| CRFD-2.5GF@VGA | Basic Res | 93.78 | 92.16 | 77.87 | 0.67 | 2.53 | 4.2 |
| SCRFD-2.5GF@VGA | Basic Res | 93.76 | 92.17 | 78.35 | 0.67 | 2.53 | 4.2 |
| SCRFD-2.5GF@Multi-Scale | Basic Res | 95.21 | 94.44 | 89.92 | 0.67 | 155.69 | - |

space. Therefore, `CRFD-2.5GF` can obviously outperform the BFBox and evolutionary method by $1.96\%$ and $1.75\%$ on the hard track.

**Sample redistribution.** For scale augmentation, we first manually extend the default scale set $\{0.3, 0.45, 0.6, 0.8, 1.0\}$ by adding larger scales $\{1.2, 1.4, 1.6, 1.8, 2.0\}$. By adding this hand-crafted sample redistribution, the hard set APs significantly increase by $7.15\%$ for the baseline and $6.5\%$ for the proposed `CRFD`, indicating the benefit from allocating more training samples on the feature map of stride $8$. By employing the optimized scale augmentation from searching, the hard set AP further increases by $0.48\%$ for the proposed `CRFD`. For `SCRFD-2.5GF`, the best scale augmentation set searched is $\{0.5, 0.7, 0.8, 1.0, 1.1, 1.2, 1.4, 1.5, 1.8, 2.0, 2.3, 2.6\}$. As we can see from these discrete scales, faces around the original scale are preferred for training, along with an appropriate probability and ratio of zooming-out.

### 4.3 COMPUTATION REDISTRIBUTION ACROSS DIFFERENT COMPUTE REGIMES

Besides the complexity constraint of 2.5 GFlops, we also utilize the same two-step computation redistribution method to explore the network structure optimization for higher compute regimes (*e.g.* 10 GFlops and 34 GFlops) and lower compute regimes (*e.g.* 0.5 GFlops and 1.0 GFlops). In Fig. 5, we show the computation redistribution and the optimized network structures under different computation constraints.

Our final architectures have almost the same flops as the baseline networks. From these redistribution results, we can draw the following conclusions: (1) more computation is allocated in the backbone and the computation on the neck and head is compressed; (2) more capacity is reallocated in shallow stages due to the specific scale distribution on WIDER FACE; (3) for the high compute regime (*e.g.* 34 GFlops), the explored structure utilizes the bottleneck residual block and we observe significant depth scaling, instead of width scaling in shallow stages. Scaling the width is subject to over-fitting due to the larger increase in parameters (Bello et al., 2021). By contrast, scaling the depth, especially in the earlier layers, introduces fewer parameters compared to scaling the width; (4) for the mobile regime (0.5 GFlops), allocating the limited capacity in the deep stage (*e.g.* C5) for the discriminative features captured in the deep stage, can benefit the shallow small face detection by the top-down neck pathway.

Table 3: Accuracy and efficiency of different light-weight models on the WIDER FACE validation set. #Params and #Flops denote the number of parameters and multiply-adds. "Infer" refers to network inference latency on NVIDIA 2080TI.

| Method | Backbone | Easy | Medium | Hard | #Params(M) | #Flops(G) | Infer(ms) |
|---|---|---|---|---|---|---|---|
| RetinaFace@VGA | MobileNet0.25 | 87.78 | 81.16 | 47.32 | 0.44 | 0.802 | 7.9 |
| RetinaFace@Original | MobileNet0.25 | 89.58 | 87.11 | 69.12 | 0.44 | 2.358 | - |
| RetinaFace@Multi-Scale | MobileNet0.25 | 91.4 | 89.2 | 82.5 | 0.44 | 49.28 | - |
| FaceBoxes@VGA | - | 76.17 | 57.17 | 24.18 | 1.01 | 0.275 | 2.5 |
| FaceBoxes@Original | - | 84.5 | 77.7 | 40.4 | 1.01 | 0.809 | - |
| FaceBoxes@Multi-Scale | - | 85.9 | 81.6 | 55.7 | 1.01 | 16.93 | - |
| libfacedetection@Original | - | 85.6 | 84.2 | 72.7 | 2.33 | 3.25 | - |
| LFFD@Original | - | 91.0 | 88.0 | 77.8 | 2.15 | 27.20 | - |
| MobileNet-1.0GF@VGA | MobileNet0.25 | 91.66 | 89.28 | 70.46 | 0.63 | 1.024 | 4.9 |
| CRFD-1.0GF@VGA | Depth-wise Conv | 92.38 | 90.57 | 74.80 | 0.64 | 0.982 | 4.1 |
| SCRFD-1.0GF@VGA | Depth-wise Conv | 92.36 | 90.58 | 76.03 | 0.64 | 0.982 | 4.1 |
| SCRFD-1.0GF@Original | Depth-wise Conv | 91.89 | 89.96 | 84.70 | 0.64 | 2.89 | - |
| SCRFD-1.0GF@Multi-Scale | Depth-wise Conv | 93.87 | 92.99 | 88.74 | 0.64 | 60.39 | - |
| MobileNet-0.5GF@VGA | MobileNet0.25 | 90.38 | 87.05 | 66.68 | 0.37 | 0.507 | 3.7 |
| CRFD-0.5GF@VGA | Depth-wise Conv | 90.57 | 88.12 | 68.51 | 0.57 | 0.508 | 3.6 |
| SCRFD-0.5GF@VGA | Depth-wise Conv | 90.80 | 88.43 | 68.82 | 0.57 | 0.508 | 3.6 |
| SCRFD-0.5GF@Original | Depth-wise Conv | 90.35 | 88.21 | 81.46 | 0.57 | 1.49 | - |
| SCRFD-0.5GF@Multi-Scale | Depth-wise Conv | 92.71 | 91.45 | 86.23 | 0.57 | 31.24 | - |

## 4.4 ACCURACY AND EFFICIENCY COMPARISONS ON WIDER FACE

As shown in Tab. 2 and Tab. 3, we compared the proposed SCRFD with other state-of-the-art face detection algorithms (*e.g.* DSFD (Li et al., 2019), RetinaFace (Deng et al., 2020b), BFBox (Liu & Tang, 2020), HAMBox (Liu et al., 2020) and TinaFace (Zhu et al., 2020)) as well as light-weight face methods (*e.g.* Faceboxes (Zhang et al., 2017a), libfacedetection (Feng et al., 2021) and LFFD (He et al., 2019b)). Overall, all of the proposed SCRFD models provide considerable improvements compared to the hand-crafted baseline models (*e.g.* ResNet-2.5GF and MobileNet-0.5GF), by optimizing the network structure as well as the scale augmentation, across a wide range of compute regimes.

When we fix the testing scale at $640$ as in Tab. 2, the proposed SCRFD-34GF outperforms all these state-of-the-art methods on the three subsets, especially for the hard track, which contains a large number of tiny faces. More specifically, SCRFD-34GF surpasses TinaFace by $4.78\%$ while being more than $3\times$ *faster* on GPUs. In addition, the computation cost of SCRFD-34GF is only around $20\%$ of TinaFace. As SCRFD-34GF scales the depth in the earlier layers, it also introduces fewer parameters, resulting in a much smaller model size ($9.80M$). Compared to the hand-crafted baseline (ResNet-34GF), the proposed computation redistribution and sample redistribution improve the AP by $1.27\%$ and $0.92\%$, indicating the superiority of SCRFD over manual designs. Compared to the single path one-shot NAS method, SCRFD-34GF outperforms BFBox by $15.81\%$, while using a more compact model size. As the search space of BFBox is complex, there exists a large number of low-performance architectures. In addition, BFBox only searches the backbone and neck without considering the optimization on the head. For multi-scale testing, SCRFD-34GF slightly outperforms TinaFace but consumes much less computation. For the low-compute regimes in Tab. 3, SCRFD-0.5GF significantly outperforms RetinaFace-MobileNet0.25 by $21.19\%$ on the hard AP, while consuming only $63.34\%$ computation and $45.57\%$ inference time under the VGA resolution. When the evaluation is conducted on the original image, SCRFD-0.5GF surpasses LFFD by $3.6\%$ on the hard AP, while consuming only $5.5\%$ flops.

## 5 CONCLUSIONS

In this work, we present a sample and computation redistribution paradigm for efficient face detection. Our results show significantly improved accuracy and efficiency trade-off by the proposed SCRFD across a wide range of compute regimes, when compared to the current state-of-the-art.

**Acknowledgements.** We would like to thank Hui Ni from Tencent for preparing the mobile demo of SCRFD https://github.com/nihui/ncnn-android-scrfd. Stefanos Zafeiriou acknowledges support from the EPSRC Fellowship DEFORM (EP/S010203/1), FACER2VM (EP/N007743/1) and a Google Faculty Fellowship.

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

# A APPENDIX

## A.1 TINAFACE REVISITED

Based on RetinaNet (Lin et al., 2017a), TinaFace (Zhu et al., 2020) employs ResNet-50 (He et al., 2016) as backbone, and Feature Pyramid Network (FPN) (Lin et al., 2017a) as neck to construct the feature extractor. For the head design, TinaFace first uses a feature enhancement module on each feature pyramid to learn surrounding context through different receptive fields in the inception block (Szegedy et al., 2015). Then, four consecutive $3 \times 3$ convolutional layers are appended on each feature pyramid. Focal loss (Lin et al., 2017b) is used for the classification branch, DIoU loss (Zheng et al., 2020) for the box regression branch and cross-entropy loss for the IoU prediction branch.

To detect tiny faces, TinaFace tiles anchors of three different scales, over each level of the FPN (*i.e.* $\{2^{4/3}, 2^{5/3}, 2^{6/3}\} \times \{4, 8, 16, 32, 64, 128\}$, from level $P_2$ to $P_7$). The aspect ratio is set as 1.3. During training, square patches are cropped from the original image and resized to $640 \times 640$, using a scaling factor randomly sampled from $[0.3, 0.45, 0.6, 0.8, 1.0]$, multiplied by the length of the original image's short edge. During testing, TinaFace employs single scale testing, when the short and long edges of the image do not surpass $[1100, 1650]$. Otherwise, it employs with short edge scaling at $[500, 800, 1100, 1400, 1700]$, shift with directions $[(0, 0), (0, 1), (1, 0), (1, 1)]$ and horizontal flip.

As shown in Fig. 6(a) and Tab. 4, we compare the performance of TinaFace under different testing scales. For the multi-scale testing, TinaFace achieves an impressive AP of $93.4\%$, which is the current best performance on the WIDER FACE leader-board. For large single-scale testing (1650), the AP slightly drops at $93.0\%$ but the computation significantly decreases to 1021.82 GFlops. On the original scale (1024), the performance of TinaFace is still very high, obtaining an AP of $91.4\%$ with 508.47 GFlops. Moreover, when the testing scale decreases to VGA level (640), the AP significantly reduces to $81.4\%$, with the computation further decreasing at 172.95 GFlops.

In Fig. 6(b), we illustrate the computation distribution of TinaFace on the backbone, neck and head components with a testing scale of 640. From the view of different scales of the feature pyramid, the majority of the computational costs (about $68\%$) are from stride 4, as the resolution of feature map is quite large ($120 \times 160$). From the view of the different components of the face detector, most of the computational costs (about $79\%$) are from the head, since the backbone structure is directly borrowed from the ImageNet classification task (Deng et al., 2009), without any modification.

Even though TinaFace achieves state-of-the-art performance on tiny face detection, the heavy computational cost renders it unsuitable for real-time applications.

## A.2 DETAILS OF EVOLUTIONARY BASELINE

To compare the proposed SCRFD with the other network search methods in Tab. 1, we design the evolutionary baseline (Real et al., 2019) as follows:

1. A population of networks $\mathbf{P}$ are randomly initialized. We set $|\mathbf{P}| = 50$.

2. Each network architecture from $\mathbf{P}$ is trained on the WIDER FACE training data and then the APs on the WIDER FACE validation dataset are tested.

3. Architectures with top performance are selected from $\mathbf{P}$ as parents $\mathcal{P}$. To generate child networks $\mathbf{C}$, we employ the mutation and crossover policies. Here, we set $|\mathcal{P}| = 10$ and $|\mathbf{C}| = 50$.

4. Each network architecture from $\mathbf{C}$ is trained on the WIDER FACE training data and then the APs on the WIDER FACE validation dataset are calculated.

5. The worst 50 individuals from the populations of $\mathbf{P} \cup \mathbf{C}$ are dropped and then we get the new evolutionary population $\mathbf{P}$.

6. We repeat steps 3, 4 and 5 for 20 times, resulting in 1000 network architectures as well as their validation APs. The architecture with the highest AP is selected as the final result.

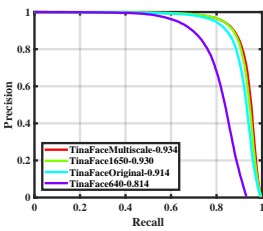 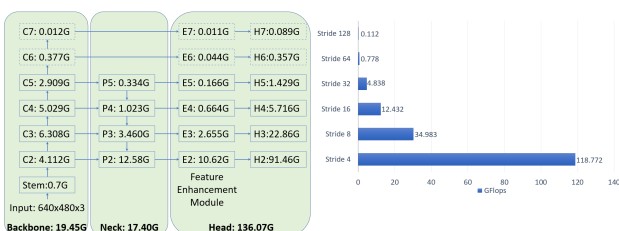

(a) Different Testing Scales          (b) Computation Distribution of TinaFace@$640 \times 480$

Figure 6: (a) Precision-recall curves of TinaFace-ResNet50 on the WIDER FACE hard validation subset, under different testing scales. (b) Computation distribution of TinaFace on backbone, neck and head with $640 \times 480$ as the testing scale.

## A.3  ALGORITHM OF COMPUTATION REDISTRIBUTION

In Algorithm 1, we show the details of the proposed two-step computation redistribution method.

---
**Algorithm 1:** Search algorithm for computation redistribution

---
**Input:** Constraint of computation cost $Y$ (in GFlops); Number of random network
  architectures $N$; Dataset for training $D_{train}$ and validation $D_{val}$; Evaluation metric $AP$.
**Output:** Best architecture $A^*$
Initialize the architecture set $\mathbb{A} = \emptyset$
**while** *length($\mathbb{A}$) < N* **do**
  $net = RandomSampling(\{d_i, w_i\})$ ;
    /* $d_i$ and $w_i$ denote block number and channel number, $i = 2, 3, 4, 5$. */
  **if** *net.Flops $\leq 1.02 * Y$ And net.Flops $\geq 0.98 * Y$* **then**
    $\mathbb{A}$.Append(net)
  **end**
**end**
$ParallelTrain(\mathbb{A}, D_{train})$
$CR1 = Bootstrap(\mathbb{A}, APs) \mid APs = Evaluate(\mathbb{A}, D_{val})$
Initialize the architecture set $\mathbb{A} = \emptyset$
**while** *length($\mathbb{A}$) < N* **do**
  $net = RandomSampling(\{CR1, n, m, h\})$ ;
    /* $n, m,$ and $h$ denote channel in neck, block and channel in head. */
  **if** *net.Flops $\leq 1.02 * Y$ And net.Flops $\geq 0.98 * Y$* **then**
    $\mathbb{A}$.Append(net)
  **end**
**end**
$ParallelTrain(\mathbb{A}, D_{train})$
$CR2 = Bootstrap(\mathbb{A}, APs) \mid APs = Evaluate(\mathbb{A}, D_{val})$
**Output:** Best architecture $A^* = choose\_top1(CR2, \mathbb{A})$.

---

## A.4  DETAILED NETWORK CONFIGURATIONS

In Tab. 5, we give the detailed network configurations for baselines and the proposed CRFD across different compute regimes.

## A.5  STATISTICS AFTER SAMPLE REDISTRIBUTION

As illustrated in Fig. 7(a), there are more faces below the scale of 32 after the proposed automatic scale augmentation strategy is used. Moreover, even though there will be more extremely tiny faces (*e.g.* $< 4 \times 4$) under the proposed scale augmentation, these ground-truth faces will be neglected during training due to unsuccessful anchor matching. As shown in Fig. 7(b), positive anchors within one epoch significantly increase at the scale of 16 and 32. With more training samples redistributed to the small scale, the branch to detect tiny faces can be trained more adequately.

Table 4: Performance and computation comparisons of TinaFace under different testing scales. The average scale of original images is around $882 \times 1024$.

| Testing Scale | AP | #Flops(G) |
|---|---|---|
| Multi-scale | 0.934 | 42333.64 |
| 1650 | 0.930 | 1021.82 |
| Original(1024) | 0.914 | 508.47 |
| 640 | 0.814 | 172.95 |

Table 5: Detailed network configurations for baselines and the proposed CRFD across different compute regimes. Basic residual blocks are used in ResNet-2.5GF and ResNet-10GF, while bottleneck residual blocks are used in ResNet-34GF. For MobileNet-1.0GF and MobileNet-0.5GF, depth-wise convolution is used in both backbone and head.

| Name | Conv Type | Stem | Backbone Depth | Backbone Width | Neck | Head |
|---|---|---|---|---|---|---|
| ResNet-34GF | Bottleneck Res | 256 | [3,4,6,3] | [256,512,1024,2048] | 128 | [256,256] |
| CRFD-34GF | Bottleneck Res | 56 | [17,16,2,8] | [56,56,144,184] | 128 | [256,256] |
| ResNet-10GF | Basic Res | 32 | [3,4,6,3] | [32,64,128,256] | 128 | [160,160] |
| CRFD-10GF | Basic Res | 56 | [3,4,2,3] | [56,88,88,224] | 56 | [80,80,80] |
| ResNet-2.5GF | Basic Res | 16 | [3,4,6,3] | [16,32,64,128] | 48 | [96,96] |
| CRFD-2.5GF | Basic Res | 24 | [3,5,3,2] | [24,48,48,80] | 24 | [64,64] |
| MobileNet-1.0GF | Depth-wise Conv | 16 | [3,3,7,3] | [32,64,128,256] | 64 | [128,128] |
| CRFD-1.0GF | Depth-wise Conv | 48 | [3,2,1,5] | [48,160,216,312] | 24 | [96,96] |
| MobileNet-0.5GF | Depth-wise Conv | 16 | [2,2,6,3] | [32,64,128,256] | 32 | [80,80] |
| CRFD-0.5GF | Depth-wise Conv | 16 | [2,3,2,6] | [40,72,152,288] | 16 | [64,64] |

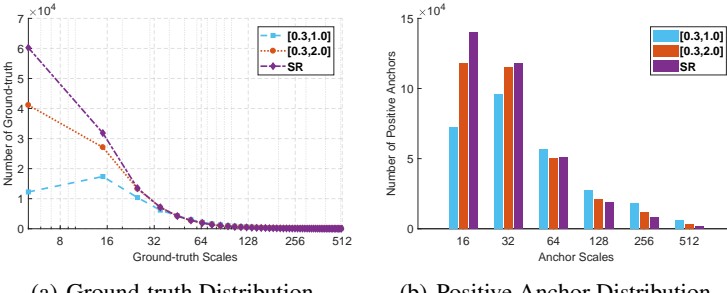

(a) Ground-truth Distribution  (b) Positive Anchor Distribution

Figure 7: Ground-truth and positive anchor distribution within one epoch for the SCRFD-2.5GF training. The baseline method employs a scale augmentation based on the hand-crafted set $[0.3, 1.0]$ and $[0.3, 2.0]$, while our method uses a searched scale set for optimized scale augmentation. The number of small faces ($< 32 \times 32$) significantly increases after the automatic scale augmentation strategy is used.

Table 6: Performance comparisons between different models on AFW, PASCAL, and FDDB datasets. The proposed SCRFD is tested on the single-scale VGA resolution.

| Methods | AFW | PASCAL | FDDB |
|---|---|---|---|
| BFBox (Liu & Tang, 2020) | 99.68 | 99.43 | 98.9 |
| HAMBox (Liu et al., 2020) | 99.90 | 99.50 | 99.10 |
| SCRFD-34GF | 99.945 | 99.597 | 99.25 |
| SCRFD-10GF | 99.900 | 99.461 | 99.07 |
| SCRFD-2.5GF | 99.821 | 98.911 | 99.02 |
| SCRFD-1.0GF | 99.696 | 98.601 | 98.69 |
| SCRFD-0.5GF | 98.603 | 98.537 | 98.14 |

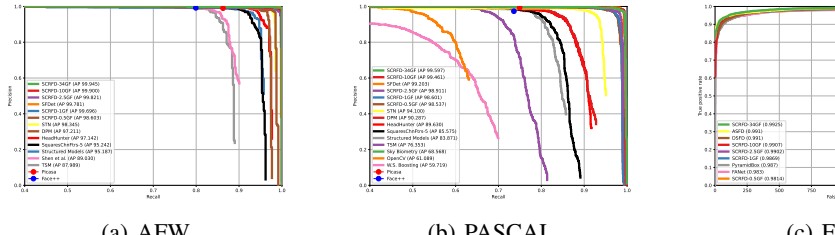

(a) AFW        (b) PASCAL        (c) FDDB

Figure 8: Precision-recall curves on AFW, PASCAL, and FDDB datasets. The proposed SCRFD is tested on the single-scale VGA resolution.

## A.6 Datasets

**WIDER FACE** The WIDER FACE dataset (Yang et al., 2016) consists of $32,203$ images and $393,703$ face bounding boxes with a high degree of variability in scale, pose, expression, occlusion and illumination. The WIDER FACE dataset is split into training ($40\%$), validation ($10\%$) and testing ($50\%$) subsets by randomly sampling from $61$ scene categories. Based on the detection rate of EdgeBox (Zitnick & Dollár, 2014), three levels of difficulty (*i.e.* Easy, Medium and Hard) are defined by incrementally incorporating hard samples.

**AFW** The AFW dataset (Zhu & Ramanan, 2012) contains $205$ high-resolution images with $473$ faces (Mathias et al., 2014) collected from Flickr. Images in this dataset contain cluttered backgrounds with large variations in viewpoint.

**PASCAL** The PASCAL face dataset (Mathias et al., 2014) is collected from the PASCAL 2012 person layout subset, includes $1,335$ labeled faces in $851$ images with large facial appearance and pose variations (*e.g.* large in-plane rotation).

**FDDB** The FDDB dataset (Jain & Learned-Miller, 2010) is a collection of labeled faces from Faces in the Wild dataset. It contains a total of $5,171$ face annotations on $2,845$ images. The dataset incorporates a range of challenges, including difficult pose angles, out-of-focus faces and low-resolution.

## A.7 Cross Dataset Evaluation and Visualization

Besides the evaluation on the WIDER FACE (Yang et al., 2016) data set, we also conduct cross dataset evaluation and test the proposed SCRFD models on AFW (Zhu & Ramanan, 2012), PASCAL (Mathias et al., 2014) and FDDB (Jain & Learned-Miller, 2010), under the VGA resolution. As shown in Fig 8, SCRFD-34GF achieves $99.945\%$ AP on AFW, $99.597\%$ AP on PASCAL, and $99.25\%$ on FDDB, surpassing BFBox (Liu & Tang, 2020) and HAMBox (Liu et al., 2020). Even though the face scale distributions on these three datasets are different from WIDER FACE, the proposed SCRFD-34GF still obtains state-of-the-art performance across different datasets, showing impressive robustness of the proposed computation and sample redistribution approaches. In addition, SCRFD-2.5GF also obtains impressive performance on different datasets with much lower computation cost ($99.821\%$ AP on AFW, $98.911\%$ AP on PASCAL, and $99.02\%$ AP on FDDB).

Fig. 9 shows qualitative results generated by SCRFD-2.5GF. As can be seen, our face detector works very well in both indoor and outdoor crowded scenes under different conditions (*e.g.* appearance variations from pose, occlusion and illumination). The impressive performance across a wide range of scales indicate that SCRFD-2.5GF has a very high recall and can detect faces accurately even without large scale testing.

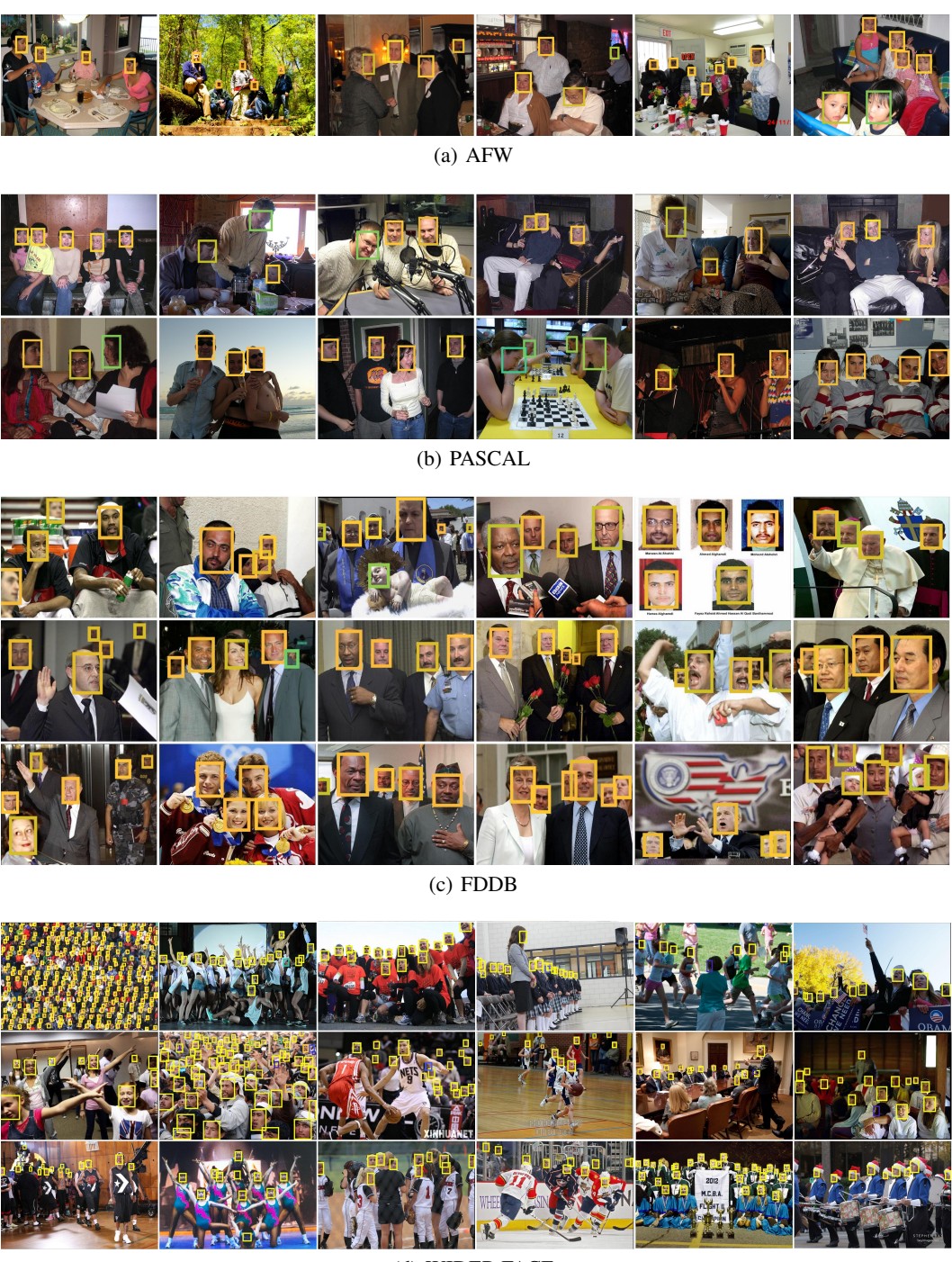

(a) AFW

(b) PASCAL

(c) FDDB

(d) WIDER FACE

Figure 9: Qualitative results on AFW, PASCAL, FDDB and WIDER FACE datasets. The proposed `SCRFD-2.5GF` is tested on the VGA resolution. Yellow rectangles show the detection results and brightness encodes the detection confidence.

