# OpenReview forum: "Sample and Computation Redistribution for Efficient Face Detection"
_ICLR.cc/2022/Conference — ICLR 2022 Poster_

### Official Review · Reviewer_zu8F · 2021-10-31

**Correctness:** 3
**Technical Novelty And Significance:** 2
**Empirical Novelty And Significance:** 3
**Recommendation:** 6
**Confidence:** 3

**Main Review:**

Strength:
•	A two-stage searching strategy is helpful to find the optimal structure in the large search space.
•	The proposed computation redistribution method is general and can be adopted by other application.
•	The scale search strategy is helpful to handle scale variations.
•	The proposed method yields the detection performance comparable to the SOTA face with lower computation cost.

Weakness
•	It lacks a figure to illustrate the overview of the whole framework.
•	An algorithm is helpful to describe the proposed computation redistribution method.
•	It needs more details in sample distribution. It is not clear how to “employ random search under the estimation metric of AP … across different computation regimes”. In addition to the purpose of shrinking the scale search space, does the binary probability have other advantage over the other probability model?
•	Cross dataset evaluation was conducted, and the overall APs were reported for AFW, Pascal, and FDDB datasets, respectively. However, it is not sufficient to claim SCRFD-2.5GF obtains SOTA performance without a comparison with the SOTA. For example, BFBox has a better performance (AP=99.43) on the Pascal dataset; HAMBox has a better performance on all these three datasets: AP=99.9 on AFW, AP=99.5 on Pascal, and 0.991 on FDDB. The authors discussed both BFBox and HAMBox in related work, but only compared with HAMBox on the WIDER FACE. In addition, Figure 5 is shown without an explanation in the text.


**Summary Of The Paper:**

This paper presents a face detection method that aims to deal with two challenges in unconstrained face detection: high computation cost and detecting small faces. Specifically, the authors adopted a network structure search method along with a two-step searching strategy for computation redistribution between the backbone, neck and head of the network. Sample redistribution between the scales is achieved using a searchable zoom-in and zoom-out space for face scale augmentation. Experiments were conducted on a benchmark dataset: WIDER FACE.

**Summary Of The Review:**

The idea of search an optimal structure considering all components to achieve both computational efficiency and performance is novel and the experimental results have supported the claim. However, details are missing in the main contributions -  searching network structure and searching scale. The search strategies implemented are rather straightforward.

---

> ### Author Response · Authors · 2021-11-23
> **Response to Reviewer zu8F**
>
> We thank the reviewer for the positive comments and detailed suggestions to improve our paper. Below, we list our replies to the questions raised.
>
> **Q1**: It lacks a figure to illustrate the overview of the whole framework.
>
> **A1**: We thank the reviewer for this suggestion. We add Fig 2 in the revised version.
> Fig 2 illustrates the computation redistribution among the backbone, neck and head. The backbone search space contains four stages, each stage having two parameters: the block number $d_i$ and block width $w_i$.
> The neck search space only includes the channel number $n$. The head is shared for the three-scale of feature maps ($N_i$), and the search space consists of the block number $m$ and channel number $h$.
>
> **Q2**: An algorithm is helpful to describe the proposed computation redistribution method.
>
> **A2**: Thanks for the suggestion. We add Algorithm 1 in the appendix of the revised version, due to the space limitation in the main paper.
>
> **Q3**: Does the binary probability have other advantage over the other probability model?  It needs more details in sample distribution.
>
> **A3**: The proposed binary probability can compress the scale search space, and it is also much easier to implement. Under the binary probability, the probability-based scale search is simplified into a discrete scale sampling from a fixed set.
> As the interval of the discrete scale set is only $0.1$, adjacent scales will have the probability of $1.0$ to approximate a higher probability around a particular scale.
>
> In this paper, we employ random search under the estimation metric of AP on WIDER FACE to construct the best scale augmentation set. More specifically, we set $s_{min} = 0.1$ and $s_{max} = 3.0$. Then, we randomly select $8$ to $20$ discrete scale values to construct each scale augmentation set and train CRFD models under $320$ different scale augmentation sets. Finally, the scale augmentation set with the highest detection performance is selected for optimized scale augmentation. Please check the supplementary code for more implementation details.
>
> **Q4**: it is not sufficient to claim SCRFD-2.5GF obtains SOTA performance on AFW, Pascal, and FDDB datasets.
>
> **A4**: Thanks for your comment.
> We move cross dataset evaluation into appendix due to the space limitation in the main paper. As shown in Fig. 8,
> SCRFD-34GF achieves $99.945\\%$ AP on AFW, $99.597\\%$ AP on PASCAL, and $99.25\\%$ on FDDB, surpassing BFBox and HAMBox. In addition, SCRFD-2.5GF also obtains impressive performance on different datasets with much lower computation cost ($99.821\\%$ AP on AFW, $98.911\\%$ AP on PASCAL, and $99.02\\%$ AP on FDDB). We add Tab. 6 to compare the APs of different models on different test sets. We have changed SCRFD-2.5GF to SCRFD-34GF to claim state-of-the-art performance on AFW, PASCAL and FDDB.
>
> |Methods     | AFW   | PASCAL| FDDB |
> | ------     | ------| ----- | -----|
> |BFBox       |99.68  | 99.43 |  98.9|
> |HAMBox      |99.90  | 99.50 | 99.10|
> |SCRFD-34GF  |99.945 | 99.597| 99.25|
> |SCRFD-10GF  |99.900 | 99.461| 99.07|
> |SCRFD-2.5GF |99.821 | 98.911| 99.02|
> |SCRFD-1.0GF |99.696 | 98.601| 98.69|
> |SCRFD-0.5GF |98.603 | 98.537| 98.14|

---

> > ### Comment · Reviewer_zu8F · 2021-11-29
> > **Response to the authors' feedback**
> >
> > The authors addressed my questions. Thanks.

---

### Official Review · Reviewer_Vs5g · 2021-11-01

**Correctness:** 4
**Technical Novelty And Significance:** 4
**Empirical Novelty And Significance:** 4
**Recommendation:** 8
**Confidence:** 3

**Details Of Ethics Concerns:**

Nil

**Main Review:**

1. The result on the wider-face dataset is very good, outperforming most of the state-of-the-art approaches.
2. The paper relied on a "meticulously designed methodology of search space optimization". Architectures are randomly sampled to find the likely range of the best architecture. In this sense, will evolutionary approach do the same trick?
3. Also, this paper is basically a kind of Neural Architecture Search, and would like to see a comparison with at least evolutionary method.
4. The choice of TinaFace as base network is straightforward. However, is it possible that after NAS, a close runner-up can be better than the champion?

**Summary Of The Paper:**

The paper uses a kind of Neural Architecture Search for finding the best computation redistribution on the base network, based on TinaFace, and the best sample redistribution to find the best network for face detection. It performs very well on the wider-face dataset, outperforming most of the state-of-the-art approaches.

**Summary Of The Review:**

The paper uses computation redistribution and sample redistribution for searching the best architecture based on TinaFace.  I'll say that there is enough innovation from the paper, and the result is very good. Although I have some questions posted in the main review, I think the paper is acceptable in the conference.

---

> ### Author Response · Authors · 2021-11-23
> **Response to Reviewer Vs5g**
>
> We thank the reviewer for the positive and detailed review, as well as the suggestions for improvement. Our responses to the reviewer’s questions are below:
>
> **Q1**: The result on the wider-face dataset is very good, outperforming most of the state-of-the-art approaches.
>
> **A1**: Thanks for your confirmation regarding the performance of the proposed method.
>
> **Q2**: Will evolutionary approach do the same trick?
>
> **A2**: As the reviewer points out, the motivation of our method and the evolutionary approach are related, since both are finding the latent best architectures based on prior knowledge. Specifically, our method utilizes an empirical bootstrap to estimate the range of best architecture candidates, while the evolutionary approach adopts mutations or crossover to generate the best architecture candidates. However, there also exists an intrinsic difference between them. Our method aims to find the best architectures by eliminating the low-quality candidate architectures from the initialize search space, thus the majority of candidates in our designed search space are acceptable. This is why we only randomly sample 320 models in our search space. On the contrary, the evolutionary approach directly finds crossover or mutated populations from the initial search space, which includes a large number of under-performing architectures. This poses a huge challenge for the evolutionary approach on generating superior crossover or mutation populations on such a large and poor search space.
>
> **Q3**: a comparison with at least evolutionary method
>
> **A3**: In Tab. 1, we add the comparison between our method and the evolutionary search method, under the constraint of 2.5 GFlops. The evolutionary approach adopts mutations and crossover, in order to gradually generate better architecture candidates from the randomly initialized search space, which contains a large number of sub-optimal architectures. We put the details of our evolutionary baseline in the Appendix. 2. Moreover, the evolutionary approach employs a sequential search strategy. By contrast, CRFD-2.5GF employs the parallel random search strategy. After we sample $320$ models, the empirical bootstrap is used to estimate the optimized computation distribution of the best-performed architecture candidates, which directly eliminates the low-quality architectures from the initialized search space.
> Therefore, CRFD-2.5GF can obviously outperform the evolutionary method by $1.75\\%$ on the hard track of the WIDER FACE dataset.
>
> **Q4**: a close runner-up can be better than the champion?
>
> **A4**: Yes. A close runner-up could potentially beat the champion when using the proposed sample and computation redistribution method.
>
> In this paper, we choose TinaFace as the baseline, as it is currently the best-performing, as well as, reproducible face detector with open-source code. In Tab. 2, HAMBox [3] is a runner-up method. More specifically, HAMBox finds that many unmatched anchors in the training phase also have strong localization ability and proposes an online high-quality anchor mining strategy. With the increase of training iteration, HAMBox is able to mine unmatched high-quality anchors for outer faces. In addition, a regression-aware focal loss based on IoU is explored to effectively weigh those new compensated high-quality anchors. HAMBox achieves $76.75\\%$ AP on the hard track of WIDER FACE. After applying the proposed sample redistribution and two-step computation redistribution under the computation constraint of 34 GFlops, the performance of HAMBox increases to $85.17\\%$, surpassing TinaFace by $3.74\\%$. This improvement indicates the effectiveness of the proposed method under different anchor matching strategies and loss designs.
> In the proposed SCRFD-34GF ($86.21\\%$), we employ Adaptive Training Sample Selection (ATSS) for positive anchor matching. In the detection head, weight sharing and Group Normalization are used. The losses of classification and regression branches are Generalized Focal Loss (GFL) and DIoU loss, respectively. These differences in the setting, enable SCRFD-34GF to outperform HAMBox-34GF with the proposed sample and computation redistribution.
>
> [3] Liu, Yang, Xu Tang, Junyu Han, Jingtuo Liu, Dinger Rui, and Xiang Wu. "Hambox: Delving into mining high-quality anchors on face detection." In 2020 IEEE/CVF Conference on Computer Vision and Pattern Recognition (CVPR), pp. 13043-13051. IEEE, 2020.

---

### Official Review · Reviewer_oWUC · 2021-11-03

**Correctness:** 4
**Technical Novelty And Significance:** 3
**Empirical Novelty And Significance:** 3
**Recommendation:** 8
**Confidence:** 3

**Main Review:**

strengths:
+ Both of the proposed CR and SR are effective and the ablation study showed that they definitely improved the detection accuracy.

+ The proposed method is effective especially for detection of small faces and the experimental results support this (the accuracy improves largely on WIDER FACE Hard which includes many small faces).

weaknesses:
- The proposed methods are effective for detection of objects with wide range of scales, which might have limited applications.

**Summary Of The Paper:**

The authors proposed face detection algorithm based on the optimized network architecture and data sampling strategy.  Their novelties are two-fold: one is Computation Redistribution (CR) which optimally reallocates the computation between the backbone, neck and head of the model, and another is Sample Redistribution (SR) which automatically redistributes more training samples for shallow stages.  The ablation study showed that both of proposed methods are effective and the comparative study showed that their whole pipeline achieved the highest accuracy among state-of-the-art methods on the public WIDER FACE dataset.

**Summary Of The Review:**

Though the proposed methods might be fruitful only for object detection of wider range of object scales, the experimental evaluations showed that they successfully improved the detection accuracy for small faces.

---

> ### Author Response · Authors · 2021-11-23
> **Response to Reviewer oWUC**
>
> We thank the reviewer for the time put on reviewing our paper and the positive feedback that helps us improve our paper.
>
> **Q1**: The proposed methods might be fruitful only for object detection of wider range of object scales.
>
> **A1**: In this paper, the proposed computation redistribution and sample redistribution methods are mainly designed for detecting tiny face instances in low-resolution images (VGA resolution). The proposed method can also provide useful insights
> to efficient object detection and object segmentation,
> since they share many similarities to our problem. In addition, our work can be insightful for general classifier design, as our computation redistribution over the different stages of the backbone, is similar to the backbone design of CNN-based classifiers.

---

> > ### Comment · Reviewer_oWUC · 2021-11-30
> > **Response to author's feedback**
> >
> > I understand it.  Thank you for helpful comments.

---

### Official Review · Reviewer_prPQ · 2021-11-16

**Correctness:** 3
**Technical Novelty And Significance:** 2
**Empirical Novelty And Significance:** 3
**Recommendation:** 6
**Confidence:** 3

**Main Review:**

* Strength
  * The paper indicates that most of the faces (78.93%) in WIDER FACE are smaller than 32×32 pixels. Under this specific scale distribution, both network structure and scale augmentation need to be optimized. The experiments show that the SCRFD indeed enhances the
detection performance on small faces.
  * Both of the proposed CR and SR are effective and achieve great improvements on well-known datasets. The author provided a detailed ablation study that showed that they improved the detection accuracy than the baseline.

* Weakness
  * The search strategies(CR) implemented are rather straightforward and not interesting.
  * The experiments mainly focus on comparison with some general detectors. There is almost no comparison experiment with some well-known network search methods. So I am not convinced whether the improvements are mainly due to network search or your contribution.



**Summary Of The Paper:**

The author is motivated by two simple but effective methods: 1) Computation Redistribution (CR) which reallocates the computation between the backbone, neck and head calculation; 2) Sample Redistribution (SR) augments training samples for the most needed stages.  The author uses a simplified search space for computation redistribution across different components and designs a searchable zoom-in and zoom-out space for face-specific scale augmentation. The SCRFD-34GF yields state-of-the-art performances in many datasets(e.g. WIDER FACE).

**Summary Of The Review:**

* The methods of searching the network structure consider each component to achieve both performance and latency.
* Comparison with other SOTA network search methods is inadequate but necessary.
* Although the method is straightforward and not very interesting, it indeed gets great improvements than the baseline.

---

> ### Author Response · Authors · 2021-11-23
> **Response to Reviewer prPQ**
>
> We thank the reviewer for the positive review and the valuable feedback to improve our paper. Answers to specific points are below:
>
> **Q1**: Although the method is straightforward and not very interesting, it indeed gets great improvements than the baseline.
>
> **A1**: The proposed SCRFD is a simple, yet effective method, to optimize model performance under a computational cost constraint. SCRFD is practical and of interest for a lot of real-world applications, such as face detection on the mobile devices. SCRFD is also interesting and insightful for efficient network designing, as we achieve impressive performance by the suggested search space simplification and redistribution of the computational power to the most needed parts.
>
> **Q2**: Comparison with other SOTA network search methods is inadequate but necessary.
>
> **A2**: In Tab. 1, we add a comparison between our method and the single path one-shot NAS method (BFBox [1]), as well as the evolutionary search method [2], under the constraint of 2.5 GFlops. BFBox aims to design a face-appropriate search space by combing some excellent block designs, such as bottleneck block, densenet block, shufflenet block, inceptionv4 block and some heuristic blocks. However, such a combination mode generates a complex and redundant search space, which inevitably involves a vast body of terrible candidate architectures. The evolutionary approach adopts mutations and crossover to gradually generate better architecture candidates from a randomly initialized search space, which also contains a large number of under-performing architectures. We have included the details of our evolutionary baseline in Appendix 2. The evolutionary approach employs a sequential search strategy. Our approach, CRFD-2.5GF, employs the parallel random search strategy. After we get $320$ models, empirical bootstrap is used to estimate the optimized computation redistribution of the best-performing architecture candidates, which directly eliminates the low-quality architectures from the initialized search space.
> Therefore, CRFD-2.5GF can obviously outperform the BFBox and evolutionary method by $1.96\\%$ and $1.75\\%$ on the hard track of the WIDER FACE dataset.
>
> In Tab. 2, we also add the comparison between SCRFD-34GF and the single path one-shot NAS method (BFBox). Under the test resolution of $640$, SCRFD-34GF outperforms BFBox by $15.81\\%$, while using a more compact model size. As the search space of BFBox is complex, there exists a large number of inefficient architectures. In addition, BFBox only searches the backbone and neck without considering the optimization on the head.
>
> We have revised our manuscript to include the above details, and have highlighted the additions with blue color.
>
> [1] Liu, Yang, and Xu Tang. "Bfbox: Searching face-appropriate backbone and feature pyramid network for face detector." In Proceedings of the IEEE/CVF Conference on Computer Vision and Pattern Recognition, pp. 13568-13577. 2020.
>
> [2] Real, Esteban, Alok Aggarwal, Yanping Huang, and Quoc V. Le. "Regularized evolution for image classifier architecture search." In Proceedings of the aaai conference on artificial intelligence, vol. 33, no. 01, pp. 4780-4789. 2019.

---

> > ### Comment · Reviewer_prPQ · 2021-11-30
> > **Response to the authors' feedback**
> >
> > The author has solved my problem.

---

### Decision · Program_Chairs · 2022-01-20

**Decision:**

Accept (Poster)

**Comment:**

This paper received 4 quality reviews, with the final rating of 8 by 2 reviewers, and 6 by the other 2 reviewers. All reviews recognize the contributions of this work, especially its superior performance. The AC concurs with these contributions and recommends acceptance.